# Self Reward Design with Fine-grained Interpretability

## Abstract

Transparency and fairness issues in Deep Reinforcement Learning may stem from the black-box nature of deep neural networks used to learn its policy, value functions etc. This paper proposes a way to circumvent the issues through the bottom-up design of neural networks (NN) with detailed interpretability, where each neuron or layer has its own meaning and utility that corresponds to humanly understandable concept. With deliberate design, we show that lavaland problems can be solved using NN model with few parameters. Furthermore, we introduce the Self Reward Design (SRD), inspired by the Inverse Reward Design, so that our interpretable design can (1) solve the problem by pure design (although imperfectly) (2) be optimized via SRD (3) perform avoidance of unknown states by recognizing the inactivations of neurons aggregated as the activation in $w_{unknown}$.

## 1 Introduction

Reinforcement Learning (RL) and Deep Neural Network (DNN) have recently been integrated to solve problems with remarkable performance. The deep reinforcement learning greatly improves the state-of-the-art of control and, in the words of Sutton & Barto (2018), *learning from interaction*. Among the well-known successes are (1) the Deep Q-Network (Mnih et al. (2015)) which enabled machine to play Atari Games with incredible performance, and (2) AlphaGo which is capable of playing notoriously complex game of Go (Silver et al. (2016)), which has also been further developed and popularized as being capable of defeating human at pro level. Although DNN has proven to possess great potentials, it is a blackbox that is difficult to interpret. To address this difficulty, various works have emerged, thus we have a host of different approaches to eXplainable Artificial Intelligence (XAI); see surveys Arrieta et al. (2020); Gilpin et al. (2018); Tjoa & Guan (2020). They have shed some lights into the inner working of a DNN, but there may still be large gaps to fill. Note that there is no guarantee that interpretability is even attainable, especially when context-dependent interpretability can be subjective.

In this paper, we propose the Self Reward Design, a non-traditional RL solution that combines highly interpretable human-centric design and the power of DNN. Our robot (representing any artificial agent) rewards itself through purposeful design of DNN architecture, enabling it to solve problem **without training**. The solution might be sub-optimal, but the use of trainable DNN modules (we use pytorch, specifically) addresses the problem. We show that performance is improved while interpretability is retained.

This paper is arranged as the following. We start with clarifications. Then we briefly go through related works that inspire this paper. Then our interpretable design and SRD training idea are demonstrated with a 1D toy example, RobotFish. Following that, we introduce SRD on the main robot navigation problem on the 2D *lavaland* with a large focus on interpretable design, followed by SRD training and unknown avoidance. Its experimental results will be discussed in the section after, and finally, we conclude with limitations and future works.

## 2 This Paper Focuses Heavily on Interpretable Human Design

**What exactly is this paper about?** Demonstrations of how two reinforcement learning problems are solved in an interpretable manner through self-reward mechanism. Readers will find how we

design different components taylored to different problem components. See appendix A.1 for more remarks: the paper has been heavily reorganized to direct readers' focus on *interpretable design*.

**But what is interpretability?** While there may be many ways to talk about interpretability, interpretability in the context of this paper is fine-grained, i.e. we go all the way down to *directly manipulating weights and biases* of DNN modules. DNN modules are usually optimized using gradient descent from random initialization, thus the resulting weights are hard to interpret. In our SRD model, the meaning and purpose of each neural network component can be explicitly stated with respect to the environmental and model settings.

**How do we compare our interpretability** with existing explainable deep RL methods? Since we directly manipulate the weights and biases, our interpretability is at a very low level of abstraction, unlike post-hoc analysis e.g. saliency (Greydanus et al. (2018)) or semantically meaningful high level specification such as reward decomposition (Juozapaitis et al. (2019)). In other words, we aim to be as transparent as possible, allowing users to understand the model from the most basic unit.

**Baseline**. Our main result on 2D robot lavaland example achieves a high performance of approximately 90% accuracy with 10% randomness to allow for exploration. Hence, we believe comparing accuracy performance with other RL methods is irrelevant, and, most importantly, distracting from our focus of interpretability. If possible, *we want to compare the level of interpretability*. Quantitative comparison is tricky, then how do we compare? What baseline to use? We have in fact answered this question in the previous paragraph: our interpretability is fine-grained as we directly manipulate weights and biases.

**Different design for different contexts**. There is probably no cookie-cutter framework for SRD. Solution design that is interpretable according to the above definition necessarily and heavily depends on the context, i.e. the shape of the system, thus we might need different formula of losses. This paper demonstrates how purposeful and deliberate designs work on two RL problems. In our robot fish example, the task is simply to survive, hence the model rewards actions that encourages survival e.g. "eat when hungry". In 2D robot lavaland example, each tile is considered as a potential checkpoint to reach before reaching the final target, thus the model uses tile-based module that rewards the tile relative to other tiles: the preferred tile is scored higher.

## 2.1 RELATED WORKS: FROM RL TO DEEP RL TO SRD

**RL imperfect human design**. RL system can be set up by human manually specifying the rewards. Unfortunately, human design can easily be imperfect since the designers might not necessarily grasp the full extent of complex problem. For RL agents, designers' manual specification of rewards are fallible, subject to errors and problems such as *negative side effect of a misspecified reward* (Clark & Amodei (2016)) and *reward hacking* (Russell & Norvig (2010)). Dylan's *inverse reward design* (IRD) paper (Hadfield-Menell et al. (2017)) addresses this problem directly: it allows a model to learn beyond what imperfect designers specify. An important component of our solution to imperfect designer problem is the implementation of *unknown avoidance*, particularly $w_{unknown}$ in our 2D robot. Also see appendix A.2 for Reward Design Problem (RDP).

**From RL to Deep RL**. Not only is human design fallible, it might be very difficult, especially for complex problems. In the introduction, we mention that Deep RL (DRL) solves this by combining RL and the power of DNN. However, DRL rewards have been produced by the black-box: they become difficult to understand and thus explainable DRL emerges to address that problem.

RL papers that address explainability/interptretability problems have been compiled in survey papers (eg. Heuillet et al. (2021); Puiutta & Veith (2020)). Saliency, a common XAI method, has been applied to visualize deep RL's mechanism (Greydanus et al. (2018)). Relational deep RL uses a relational module that not only improves the agent's performance on StarCraft II and Box-World, but also provides visualization on the attention heads useful for interpretability (Zambaldi et al. (2019)). Other methods to improve interpretability include reward decomposition, in which each part of the decomposable reward is semantically meaningful (Juozapaitis et al. (2019)); do refer to the survey papers for several other ingenious designs and investigations into the interpretability of deep RL.

**From DRL to SRD**. Our model is called *self reward* design because our robot computes its own reward, similar to DRL computation of Q-values. However, human design is necessary to put con-

straints on how self-rewarding is performed so that interpretability is maintained. Human designer has the responsibility of understanding the problems, dividing the problems into smaller chunks and then finding the relevant modules to plug into the design in a fully interpretable way (see for example how we use convolution layer to create the *food location detector*, section 2.1).

**Computational Efficiency of SRD**. The models only use modules from standard DNN modules such as convolution (conv), deconvolution (deconv) and fully-connected (FC) layers with few trainable parameters (e.g. only 180 parameters in Robot2NN). With proper choice of initial parameters, we can skip the long, arduous training and optimization processes that are usually required for deep learning to learn unseen concepts. We trade off the time spent on training algorithm with the time spent on human design, thus addressing what Kahn et al. (2018) called *sample inefficiency* (the need for large dataset hence long training time) in a human-centric way.

## 2.2 OTHER RELEVANT CONCEPTS

**Self-supervised Learning**. DRL like Value Prediction Network (VPN, Oh et al. (2017)) is self-supervised. Exploration-based RL algorithm is applied so that data are gathered real-time for training: the model thus optimizes its reward function. Our model is similar in this aspect. Unlike VPN, however, our interpretable design avoids all the abstraction of DNN. Our SRD training is also self-supervised in the sense that we do not require datasets with ground-truth labels. Instead, we induce semantic bias via interpretable components to achieve the correct solutions. Also see appendix A.2.

**Imagination component**. Our pipeline includes several rollouts of possible future trajectories similar to existing RL papers that use imagination components, with differences as the following. Compared to Kalweit & Boedecker (2017) which optimizes towards target value $\hat{y}_i = r_i + \gamma Q'_{i+1}$ (heavily abbreviated), SRD (1) is similar because agent updates on every imaginary sample available, but (2) has different, context-dependent loss computation. Also see appendix A.2 for more references.

## 3 ROBOT FISH: 1D TOY EXAMPLE

**Problem setting**. To broadly illustrate the idea, we use a one-dimensional model Fish1D with Fish Neural Network (FishNN) deliberately designed to survive the simple environment. Robot Fish1D has *energy* which is represented by a neuron labelled $F$. Energy diminishes over time. If the energy reaches 0, it dies. The environment is $env = [e_1, e_2, e_3]$ where $e_i = 0.5$ indicates there is a food at position $i$ and no food if $e_i = 0$. The fish is always located in the first block of $env$, fig. 1(A). In this problem, 'food here' scenario is $env = [0.5, 0, 0]$ which means the food is near the fish. Similarly, 'food there' scenario is $env = [0, 0.5, 0]$ or $env = [0, 0, 0.5]$, which means the food is somewhere ahead and visible. 'No food' scenario is $env = [0, 0, 0]$.

**Fish1D's Actions**. (1) 'eat': recover energy $F$ when there is food in its current position. (2) 'move': movement to the right. In our implementation, 'move' causes $env$ to be rolled left. If we treat the environment as an infinite roll tape and $env$ as fish vision's on the 3 immediately visible blocks, then the food is available every 5 block.

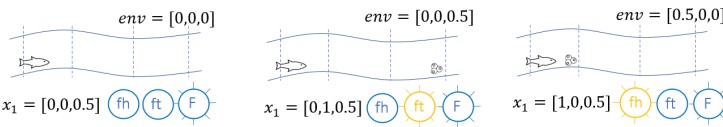

Figure 1: Robot Fish setting. The fish is half full/hungry, $F = 0.5$. Left: no food. Middle: food on $env_3$, thus "food there" neuron lights up. Right: food on $env_1$, thus "food here" neuron lights up.

**How to design an interpretable component of neural network?** First, we want the fish to be able to distinguish 3 scenarios previously defined: food here, food there and no food. Suppose we want a neuron that strongly activates when there is food nearby (name it *food here* neuron, *fh*), another neuron that strongly activates when there is a food nearby (name it *food there* neuron, *ft*), and we want to represent the no-food scenario as 'neither *fh* and *ft* respond'. How do we design a layer with two neurons with the above properties? We use 1D convolution layer and *selective activation* function as $\sigma_{sa}(x) = \epsilon/(||x||^2 + \epsilon)$, as the following.

**The *fh* and *ft* neurons**. Define the activation of *fh* neuron as $a_{fh} = \sigma_{sa}[conv_{fh}(env)]$ where $conv_{fh}$, a Conv1D with weight array $w_{fh} = [1, 0, 0]$ and bias $b_{fh} = 0.5$. When there is food near the fish, we get $y_{fh} = conv_{fh}(env) = [1, 0, 0] * [0.5, 0, 0] - 0.5 = 0$ where $*$ denotes the convolution operator, so $a_{fh} = \sigma_{sa}(y_{fh}) = 1$. This is a strong activation of neuron, because, by design, the maximum value of selective activation function is 1. We are not done yet. Similar to $a_{fh}$ above, define $a_{ft}$. The important task is to make sure that when 'there is food there but NOT HERE', $a_{ft}$ activates strongly but $a_{fh}$ does not. They are $w_{ft} = [0, 1, 1], b_{ft} = -0.5$. Together, they form the first layer called the Food Location Detector (FLD). Generally, we have used

$$a_\eta = \sigma_{sa}[conv_\eta(env)] \tag{1}$$

**Interpretable FishNN**. To construct the neural network responsible for the fish's actions (eat or move), we need one last step: connecting the neurons plus fish's internal state (energy) (altogether $[a_{fh}, a_{ft}, F]$) to the action output vector $[eat, move] \equiv [e, m]$ through FC layer, as shown in fig. 2 blue dotted box. The FC weights are chosen meaningfully e.g. 'eat when hungry and there is food' and to avoid scenarios like 'eat when there is no food'. This is interpretable through manual weight and bias setting.

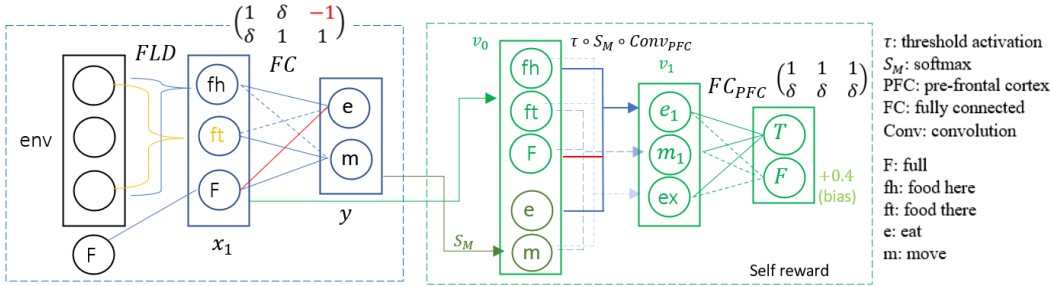

Figure 2: FishNN architecture, $\delta > 0$ a small value.

**Is FishNN's decision correct?** The prefrontal cortex (PFC) decides whether FishNN's decision is correct or not. PFC is seen in fig. 2 green dotted box. The name 'PFC' is only borrowed from the neuroscience to reflect our idea that this part of FishNN is associated with internal goals and decisions, similar to real brain (Miller et al. (2002)). How do we construct PFC? First, define *threshold activation* as $\tau(x) = Tanh(LeakyReLU(x))$. Then PFC is constructed deliberately in the same way FishNN is constructed in an interpretable way as the following.

First, aggregate states from FishNN into a vector $v_0 = [a_{fh}, a_{ft}, F, e, m]$. This will be the input to PFC. Then, $v_0$ is processed using $conv_{PFC}$ followed by softmax and threshold activation. The choice of weights and biases can be seen in table 2 in appendix B. With this design, we achieve meaningful activations $v_1 = [e_1, m_1, ex]$ as before. For example, $e_1$ is activated when "there is food and the fish eats it when it is hungry", i.e. $fh = 1, F < 1$ and $e$ is activated relative to $m_1$. The output of PFC is binary vector $[True, False] = [T, F]$ obtained from passing $v_1$ through a FC layer $FC_{PFC}$. In this implementation, we have designed the model such that the activation of any $v_1$ neuron is considered a *True* response; otherwise it is considered false. This is how PFC judges whether FishNN's action decision is correct.

**Self reward optimization**. As seen above, fish robot has FishNN that decides on an action to take and the PFC that determines whether the action is correct. Is this system already optimal? Yes, if we are only concerned with the fish's survival, since the fish will not die from hunger. However, it is not optimal with respect to average energy. We optimize the system through standard DNN backpropagation with the following self reward loss

$$loss = CEL(z, argmax(z)) \tag{2}$$

where CEL is the Cross Entropy Loss and $z = \Sigma_{i=1}^{mem}[T, F]_i$ is the accumulated decision over $mem = 8$ iterations to consider past actions (pytorch notation is used). The ground-truth used in the loss is $argmax(z)$, computed by the fish itself: hence, self-reward design.

**Results**. Human design ensures survival i.e. problem is solved correctly. Initially, fish robot decides to move rather than eat food when $F \approx 0.5$, but after SRD training, it will prefer to eat whenever

food is available, as shown in fig. 3. New equilibrium is attained: it does not affect the robot's survivability, but fish robot will now survive with higher average energy.

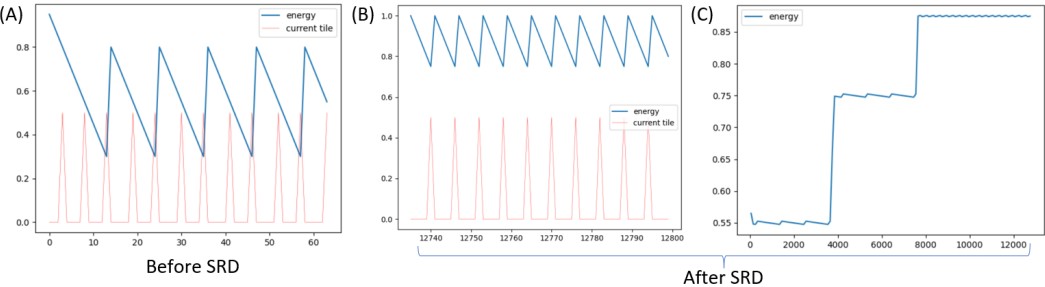

Figure 3: (A) plot of energy ($F$, blue) and food availability (red) for untrained model. SRD-trained model at its early stage looks almost identical. (B) same as (A) but after 12000 SRD training iterations. (C) Average energy of SRD-trained fish model.

## 4    2D ROBOT IN LAVALAND

**The lavaland problem**. Like Dylan's IRD paper, we use lavaland as the test bed. Given a map with either dirt (brown), grass (green), lava (red) and target (yellow) tiles and the robot's starting point (blue x), as shown in $x_{attn}$ of fig 4, the robot chooses to move either UP, DOWN, LEFT or RIGHT to reach the target. We treat lava as the "unrecognized tile" that the designer forgets to take into account (imperfect designer). Problem is considered solved if the target is reached within 36 steps. Robot should prefer not to step on grass tiles but must be designed to avoid lava tile at all cost.

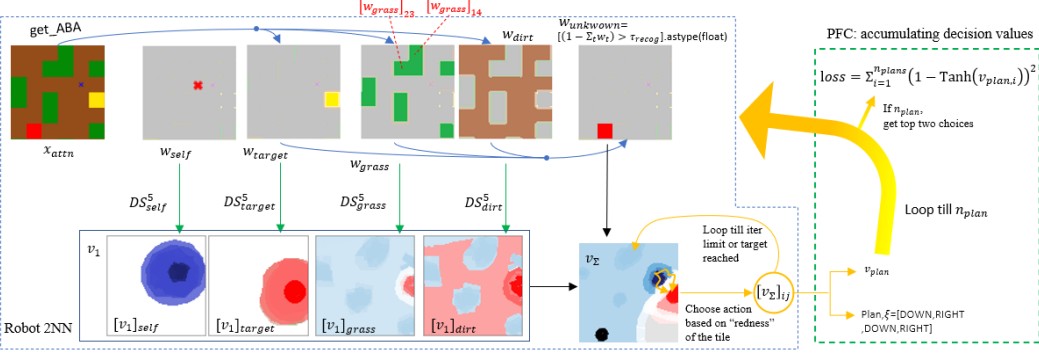

Figure 4: Robot2NN and PFC. Robot2NN makes 2D robot's action decisions. $x_{attn}$ (the whole visible map) and $w_{self}$ (the agent's position) are the direct input to the model. In $v_1, v_\Sigma$, red values are positive (desirable), white zero and blue negative (not desirable). 2D robot's PFC (green dotted box) resembles RobotFish's PFC in their function of accumulating decision values. $[w_{grass}]_{14}, [w_{grass}]_{23}$ are examples of strong activations that are interpretable through tile-based module.

**Interpretable design**. Like fish robot, we manually choose weights and bias for 2D robot's neural network to solve the problem. Convolutional layers etc are used by Robot2NN to score the tiles (red better, blue worse), i.e. robot creates a *favourability gradient*. Robot then chooses action that generally moves it from blue to red regions.

**Notations**. ABA: approximate binary array, an array whose entries are expected to be $\approx 0$ or $1$. $DS_t^n$, or DeconvSeq, is the sequence of $n$ deconvolutional layers for a tile type $t$. Deconvolutional layer, or deconv, is a regular DNN module. Each deconv is followed by Tanh activation for normalization and non-linearity. Normalization (to magnitude 1) ensures that DeconvSeq compares action choices in relative terms. Tile $t$ denotes the name of a tile, e.g. grass, but it also denotes its $[0, 1]$ normalized RGB value, e.g. for grass, $t = [0, 128, 0]/255$ or an array of tile values (they should be

obvious from the context). $\tau_{recog} = 10^{-4}$ is the recognition threshold. Designer needs to specify $P = \{p_t : t = target, grass, ...\}$. For now $P$ is the set of untrainable parameters, each $p_t$ roughly acting as the factor for scaling the true reward, inevitably affecting the robot's final preferences for or against different tiles. They induce the biases that designers input into the model in a simple, interpretable way. We also define the *unknown avoidance* parameter $u_a \geq 0$.

**Interpretable tile-based modules** are designed to explicitly map robot's response to each specific tile type. Get_ABA() function computes the ABA for each tile type: $w_t = \sigma_{sa} \circ \mu[(x_{attn} - t)^2]$ where $\mu[.]$ is the mean across RGB channel. This is reminiscent of eq. 1; the difference is, each neuron responds to a tile type at each $x_{attn}$ coordinates. Neuron activations are computed as $w_\eta$ where $\eta$ =target, grass or dirt shown in fig. 4, e.g. strong activation for grass detection occurs at $[w_{grass}]_{14}$.

**Unknown avoidance**. Like Dylan's IRD, we have a reliable mechanism for unknown avoidance, the Boolean array $w_{unknown} = [(1 - \Sigma_t w_t) > \tau_{recog}]$ to be treated as floating point numbers. From the formula, it can be seen that $w_{unknown}$ aggregates the negation of known activations. The unknown in our case is the lava tile that the imperfect human designer 'forgets' to account for.

**Interpretable tile scores**. The score $v_\Sigma$ will be used to decide what actions robot will take. It is computed as the following. First, we compute $v_1$, whose components are $[v_1]_t$, as the following. For target tiles and "self" tiles (original positions), $[v_1]_t = p_t DS_t^5[w_t]$. For any other named tiles, we also create the *favourability gradients* based on the initial and target positions, so that, for example, a grass tile nearer the target can be valued more than a dirt tile. Thus,

$$[v_1]_t = p_t DS_t^5[(w_{target} - w_{self}) * w_t] \tag{3}$$

Each deconv in DeconvSeq consists of 1 input and 1 output channel. It is fully interpretable through our manual selection of kernel weights: kernel has size 3 with center value 1 and side value 0.1 as seen in fig. 5(A). Fig. 5(B, C) show examples of trained weights. This choice of values is intended roughly to create a centred area of effect, where the center of the tile contributes most significantly to $w_t$; see for example $[v_1]_{target}$ in fig. 4. Assign $v_\Sigma \leftarrow \Sigma_t [v_1]_t$; this value will be dynamically changed throughout each plan-making process.

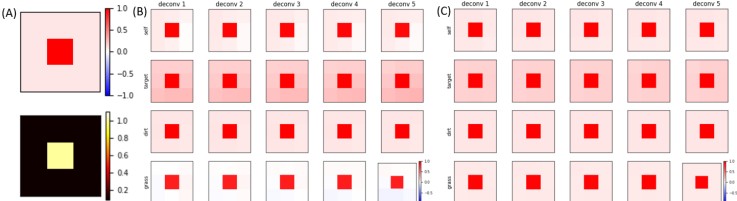

Figure 5: (A) Initial parameters; all deconvs in all DeconvSeq are initialized to the same 3x3 weights with center max value of 1 and 0.1 elsewhere. (B) Once trained, variations of weights are observed for Project A expt 1 (all 180 parameters are shown). (C) Similar to (B) but for project Compare A.

**Making one plan**. Each plan is a series of action, and each action is chosen as the following. Set the specific values for the target tile and tile of the original position to $v_0$ and $-v_0$ respectively, where $v_0 = max\{|v_\Sigma|\}$ is computed without backpropagation gradient to prevent any complications. Finally, to incorporate lava avoidance, or generally avoidance of anything previously unseen, $v_\Sigma \leftarrow v_\Sigma * (1 - w_{unknown}) + -u_a * v_0 * w_{unknown}$. From the current position, robot's neighbouring $[v_\Sigma]_i$ where $i = up, down, left, right$ values are collected, and neighbours with the top two values are chosen. From the top two choices, randomly choose one of them with a 9 to 1 odds, favouring the action with higher value. The randomness is to encourage exploration. After each action, the tile the robot leaves will be assigned $0.9v_0$ to prevent the robot from oscillating back and forth, where $v_0$ is separately computed inside step_update() function. A series of actions are chosen in this manner until the target is reached or a maximum of 36 iterations are reached. Thus we have obtained a *plan* and its score $v_{plan} = \frac{1}{N_\xi} \Sigma_{i,j \in \xi} [v_\Sigma]_{i,j}$, the mean of all values assigned to the chosen tiles where $\xi$ is the trajectory (see fig. 4, orange arrows in $v_\Sigma$).

**Imagining multiple plans for SRD optimization**. Robot makes plans by imagining $n_{plans} = 4$ different trajectories to reach the target. It plans and executes the plan with the highest $v_{plan}$. Like

Kalweit & Boedecker (2017), we use all $n_{plans}$ imagination branches for training (SRD optimization) with possibly novel loss minimization:

$$loss = \Sigma_{i=1}^{n_{plans}}(1 - Tanh(N[v_{plan,i}]))^2 \qquad (4)$$

where $N[.]$ normalizes $v_{plan}$ to the magnitude of 1, where normalization factor is computed without gradient to prevent complication. Thus, robot's PFC always undervalues its reward relative to an abstract, ideal value 1. The intended effect is to always try to maximize $v_{plan}$. A standard Stochastic Gradient Descent with learning rate $10^{-4}$ is used for optimization. This resembles fish's PFC true or false neurons used for rewarding itself, except it is continuous.

## 5 EXPERIMENT AND RESULTS

**Experimental setup and comparison**. We experiment on several initial settings as the following. For each experiment, we randomly generate and save 4096 maps to evaluate the performance of both standard design and SRD trained model, and similarly 4096 maps for SRD training. Note: All codes are included, including codes for creating animated gifs of robot traversing the lavaland. See appendix C.1 regarding multiple trials of the same experiments we conducted for reproducibility (project B, C, D etc).

**Reaching targets without training, optimized by training**. Table 1 shows that even without training ("No SRD" columns), our interpretable designs have enabled relatively high rate of problem solving. With SRD, the accuracies are further improved.

Table 1: Accuracy comparisons for Project A, project Compare A, project With-Lava-A and project Lava NOAV A. $Acc = n_{reached\ target}/4096$. "No SRD" indicates no training: reasonable accuracy is attainable purely with design. SRD training generally increases the accuracies. Starred values reflect failure modes, matching their respective histograms; see fig. 6 and 7.

| Expt no. | Project A | | Compare A | | With Lava A | | Lava NOAV A | |
|---|---|---|---|---|---|---|---|---|
| | No SRD | SRD | No SRD | SRD | No SRD | SRD | No SRD | SRD |
| 1 | 0.767 | 0.856 | 0.862 | 0.941 | 0.826 | 0.893 | 0.841 | 0.884 |
| 2 | 0.765 | 0.845 | 0.871 | 0.589* | 0.821 | 0.893 | 0.850 | 0.909 |
| 3 | 0.763 | 0.817 | 0.874 | 0.930 | 0.819 | 0.837 | 0.844 | 0.868 |
| 4 | 0.760 | 0.868 | 0.863 | 0.422* | 0.823 | 0.762* | 0.835 | 0.907 |

**Project A: standard design and SRD on maps without lava tiles**. The fraction of grass tiles is $0.3$. No robot will be spawned at the target immediately. Here, we use $p_{dirt,grass} = 0.2, -0.8$ chosen empirically. For training, we only run through 1 epoch, taking a very short time to complete (less than $0.5$ hour per project without GPU). Fig. 6(B) shows the cumulative histogram of the number of tiles traversed with $p_{target} = 2$.

**Compare A: model is more focused on the target**. This setting is similar to Project A, but we use a more extreme parameter setting, i.e. set $p_{target} = 10$. The local reward of the target tile is increased, and from table 1, we see that this generally increases the accuracies of both standard and SRD trained model (more likely to reach the target tile). The value seems to produce more unstable results as well, i.e. we empirically observe more failure modes.

**Project Lava A**. In this experiment, $0.1$ of the tiles randomly assigned as lava and unknown avoidance $u_a = 2$. We also set $p_{target} = 10$ thus, similar to *Compare A* we see higher accuracy but some failure modes as well. The model is designed to not recognize a lava tile, so a lava tile will activate the robot's Robot2NN $w_{unknown}$ at lavas' positions. Fig. 7(A) is just an example of how robot successfully avoids lava, while fig. 7(C) lava cumulative histograms show that there are nearly zero lava tiles traversed.

**Project Lava NOAV A**. Project Lava NOAV A is similar to Project Lava A, but we set $u_a = 0$, i.e. no unknown avoidance. The results are clear, we see in 7(D) that robot will traverse the lava tiles without much regards.

**Robot2NN weights and preserved interpretability**. This model is very efficient because it consists of only 180 trainable parameters, as shown fully in fig. 5(B,C). As expected of relatively simple

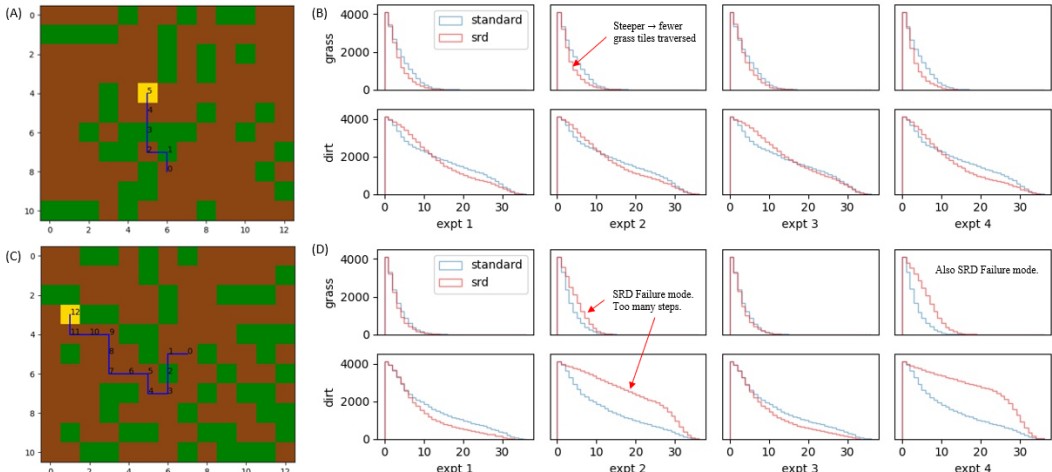

Figure 6: (A) A sample trajectory from Project A. (B) Cumulative histogram of the no. of tiles traversed by robot in Project A. Steeper histogram indicates that less tiles of the type are being traversed (for grass, steeper is better) (C) A sample trajectory from Compare A. (D) Similar to B, but for robot in Compare A.

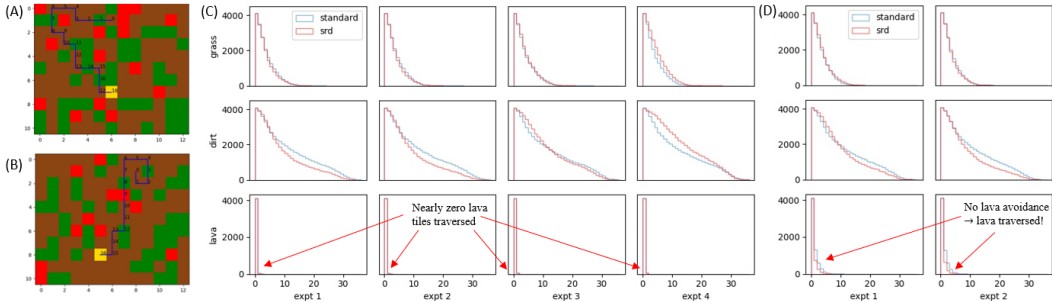

Figure 7: (A) A sample trajectory from Lava A. (B) A sample trajectory from Lava NOAV A. A lava tile is traversed. (C) Cumulative histogram of the no. of tiles traversed by robot in Lava A. Almost zero lava tiles are traversed. (D) Similar to (B) but for Lava NOAV A, in which some lava tiles are traversed.

problems, there is no need for millions of parameters required to achieve high accuracy. High performance 90% accuracy is attained, given 10% randomness is allowed. The weights of target deconv appear to have been trained towards higher positive values (redder). The center value remains the most prominent for all, thus preserving our interpretability. Looking into individual variations, fig. 5(B) shows the weights from Robot2NN model of project A expt 1 while fig. 5(C) from project Compare A expt 1. The difference in grass weights are apparent. See appendix C.2 on (1) preserved interpretability (2) why no conclusion should be drawn from this observation.

## 6    CONCLUSION, LIMITATION AND FUTURE DIRECTIONS

*Generalizability and scalability*. An important limitation to this design is the possible difficulty in creating specific design for very complex problems, for example, computer vision problems with high-dimensional state space. Problems with multitude of unknown variables might be difficult to factor into the system, or, if they are factored in, the designers' imperfect understanding of the variables may create a poor model. Future works on applying interpretable SRD on complex systems might push Deep RL community into designing systems that aim not only to attain high performance, but also to be as highly interpretable as our design. **Controlling Weights and Biases**. A

future study on regularizing weights may be interesting. While we have shown that our design provides a good consistency between the interpretable weights before and after training, it is not surprising that the combination of small differences in weights can yield different final weights that still perform well. So far, it is not clear how different parameters affect the performance of a model, or if there are any ways to regularize the training of weight patterns towards something more interpretable and consistent. So far, there is no guarantee for convergence either, and it may be difficult to provide a general proof, since the design is heavily dependent on the context. A **technical limitation** is the lack of APIs that perform the exact operations we may need to parallelize the imaginative portions of SRD optimization.

**Conclusion**. Finally, we have demonstrated interpretable neural network design with fine-grained interpretability. Human experts purposefully design models that solve specific problems based on their knowledge of the problems. SRD is introduced as a way to improve performance on top of the designers' imperfection. The system is highly interpretable with manual adjustment of weights and biases. It is also efficient since very few weights are needed compared to traditional DNN.

## ETHICS STATEMENTS

There is no specific ethical statement present. We demonstrate methods for designing neural networks for RL problem in a transparent way. We believe each part is transparently presented, and, if the paper is insufficient, the code in supp. materials should fully complements each description made in this paper.

## REPRODUCIBILITY STATEMENTS

All codes are available in the supp. materials, to be released to public repository in case of acceptance. Results are tested on different random number seeds, and irregularities or mode collapses have been presented clearly in the paper as well (see for example the annotations made on the cumulative histograms).

## ACKNOWLEDGMENTS

Anonymous now

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

# A APPENDIX

## A.1 THIS PAPER FOCUSES HEAVILY ON INTERPRETABLE HUMAN DESIGN

We reorganize this paper heavily based on ICLR 2022 reviewers' comments. Thanks to them, we understand that our focus on fine-grained interpretability (through direct manipulation of weight and biases) seems to have been lost or easily overlooked in the previous version of our paper. There is a tendency to focus on performance and solution to generalizability. We have answers for this (1) we do achieve high performance, and comparison might be very redundant. Our intention is to compare interpretability instead of performance. (2) Scalability. We naturally start by demonstrating the design on simple problems. Starting with a highly complex problems may be counterproductive and very difficult to present. In this revised paper, we have focused on the clarity of presentation.

## A.2 RELATED WORKS

In Reward Design Problem, Singh et al. (2009) observes that, given bounded agents, proxy reward function can be more optimal than the true fitness function which is distinct from the proxy. Dylan's IRD, on the other hand, approximates the true fitness function given an observed proxy reward function, assuming the designer is the bounded agent, i.e. the designer is fallible. IRD performs inversion to compute $P(w = w^*|\tilde{w}, \tilde{M})$ from $P(\tilde{w}|w^*, \tilde{M})$, where $w$ is the weight of the reward function $r(\xi; w)$, where $\xi$ is the trajectory. We cannot fit the description of our model exactly in the same language, since we opt for an interpretable design with non-traditional RL reward. The closest we have to a reward is $v_{plan}$ in Robot2NN, shown in fig. 4 indirectly as the values summed through orange arrows in $v_\Sigma$ described in the next paragraph.

**Comparison with standard Deep RL**. In the main text, we mention that our we use non-traditional reward function. Deep Q-learning loss function $L_i(\theta_i)$ computes differences between current $Q$ value (value of the current state-action pair) and its temporal difference (TD)[1] given by $r + \gamma max_{a'}Q(s', a', \theta_{i-1})$. The reward $r$ can be fixed by the designer; this can be seen on OpenAI's official github[2], and the values are then used to update Q, an expected value of total future rewards, technically also a reward. SRD is similar to DQN in this sense: there is a future cumulative reward. In turn, DQN is similar to standard RL except Q is predicted by the neural network. However, SRD does not apply the loss function $L_i(\theta_i)$ in DQN since we do not specify TD, which is dictated by the reward that explicitly adds into the sum of future reward. As mentioned, the reward value $r$ in DQN is specified by the designer and thus DQN robot aims to maximize values whose absolute magnitude of worth is known. By contrast, SRD lets the robot measure the worth of every series of actions. Since the target Tanh score is 1, every measurement of worth is relative with respect to the learning process. Even then, when the robot compares the multiple plans it makes at once, it has its own yardstick for rewarding its own decisions. The absolute magnitude of reward based on this yardstick is decided by the robot's own neural network, thus the "self reward". Finally, our robot self reward works by always undervaluing its perceived reward w.r.t the ideal plan with score 1, as mentioned in the main text.

The following supplements the main text section 2.2.

**Self-supervision**. Kahn et al. (2018) demonstrates self-supervised DRL on a simulated car that learns from real-time experience; the paper also applies the model to a real-world RC car. After hybrid graph-based RL model is incorporated, it travels around avoiding collisions without human intervention. Our robots self-supervise similarly. After the initial interpretable design, 2D Robot moves towards the target, favoring or avoiding different types of tiles without human supervision. In particular, grass tiles are recognizable, but less desirable, while lava tiles are "not recognized", hence must be completely avoided. Informally, the robots have their own preferences on how to assign the eventual values of the actions with known local consequence, while they are careful about their own ignorance induced by imperfect design (unknown avoidance).

**Imagination components**. Unlike the rollout in Racanière et al. (2017), each of our SRD rollout consists of a series of asymmetric binary choices $a_1, a_2$ chosen from $\{a \in \mathcal{A}\}$ so that $v(a_1) \geq v(a_2)$,

---

[1]see for example https://www.tensorflow.org/agents/tutorials/0_intro_rl
[2]https://github.com/openai/gym/blob/master/gym/envs/classic_control/cartpole.py

where $\mathcal{A}$ is the set of actions and $v$ is any generic function that gives each action a local value. The values will be aggregated into a self-reward. Dreamer (Hafner et al. (2020)) solves RL problem using only latent imagination where many models (such as reward, action and value models) are specified as probability distributions. By contrast, SRD creates no specific sub-model. All values are just NN activations, and they will be aggregated into impromptu, just-in-time scores, based on which the plans can be greedily chosen.

## B  ROBOT FISH

Table 2: Weights and biases in Robot Fish design in main text section 3.

| Output | Conv weights | bias |
|--------|--------------|------|
| $a_{fh}$ | $[1, 0, 0]$ | $-0.5$ |
| $a_{ft}$ | $[0, 1, 1]$ | $-0.5$ |
| $e_1$ | $[1, 0, -1, 1, 0]$ | None |
| $m_1$ | $[0, 1, -1, 0, 1]$ | None |
| $ex$ | $[-1,-1,0,0,1]$ | None |

## C  2D ROBOT

### C.1  MULTIPLE TRIALS OF THE SAME EXPERIMENTS

We perform multiple trials of each experiment for reproducibility and consistency. The main text only reports the first trial for each experimental setting since the result is generally representative. Each trial also consists of 4 experiments. Other results are in the supplementary materials (full version will be released later).

Repeat trials for Project A: project B, C and D. Histogram for project B can be seen at fig. 8.

Repeat trials for Compare A: compare B, C and D.

Repeat trials for Lava A: Lava B, C, D.

Repeat trials for Lava NOAV A: Lava NOAV B, C, D.

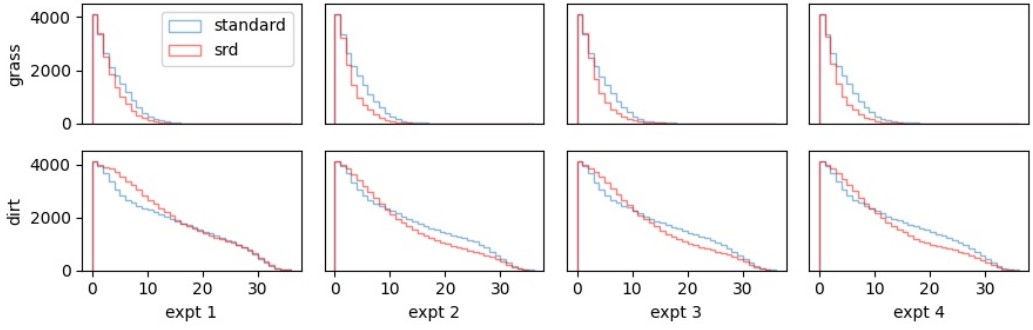

Figure 8: Project B cumulative histograms.

### C.2  DECONV WEIGHTS

**(1) Preserved interpretability**. We see from fig. C.2 that the trained models still have the general interpretable shape we initiated it with. While there is no theoretical proof, there may be an intuitive reason. Due to our interpretable design, the model starts off with a reasonable ability to solve the problem. This probably means weights and biases already reside in a high dimensional parameter

space somewhere around one of the local minima, and this local minimum is interesting because it is more interpretable i.e. weights have recognizable shapes as we have initiated. As a result, a short training leads it nearer to that local minimum, hence the overall shape of the model remains similar to the initial shape, thus interpretable.

**(2) Why no conclusion should be drawn from this observation**? It seems that project A with smaller $p_{target}$ results in relatively less preference for the grass tiles, which in turn leads to negative values (blue) for deconv for grass tiles. By comparison, project Compare A seems to have no negative values for grass tiles deconv. Unfortunately, the weights are shown only for demonstration; *there is no definite conclusion that can be drawn*. This is because other experiments similar to project A also can result in all positive deconv weights with different patterns. They still yield high accuracy, thus possible variations within even this small set of parameters can still produce similar performance. Other results are shown in the appendix. Lava A project does not yield particularly distinct patterns. We see that even failure modes can yield weights profile that look similar to non-failure modes. Further investigations may be necessary

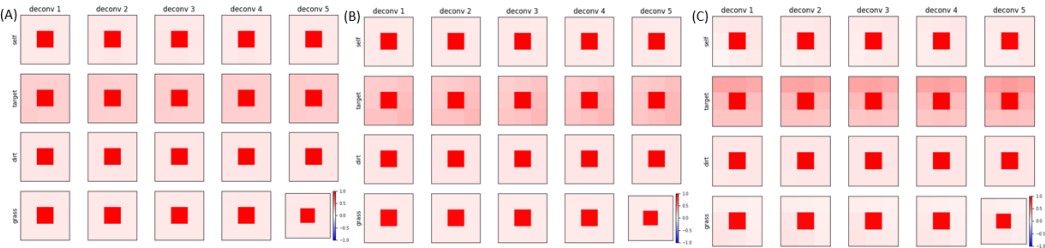

Figure 9: Weights for (A) Project Lava A expt 1 (B) Project Lava NOAV A expt 1 (C) Lava A expt 4.

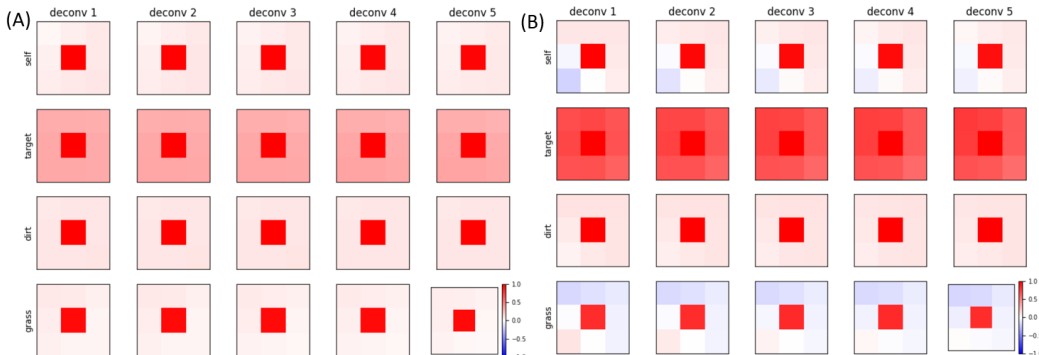

Figure 10: Weights for failure modes (A) Project Compare A expt 2 (B) Project Compare A expt 4. (A) still shows standard-looking weights.

