# OpenReview forum: "Self Reward Design with Fine-grained Interpretability"
_ICLR.cc/2022/Conference — ICLR 2022 Submitted_

### Official Review · Reviewer_cYQC · 2021-10-27

**Correctness:** 2
**Technical Novelty And Significance:** 1
**Empirical Novelty And Significance:** 2
**Recommendation:** 1
**Confidence:** 3

**Main Review:**

This paper aims to address the important problem of building intepretable neural networks. Unfortunately, it is poorly written and I found it difficult to follow.

The general form of the SRD method (independent of the task) is not explained in the paper. The method is illustrated for two environments and seems to be different for each environment, e.g. proposing two different loss functions that both seem ad-hoc and not well-motivated. The explanation of how it works is not clear, e.g. I didn't understand what sentences like this mean: "the prefrontal cortex (PFC) modules judge the worth of the fish’s own action against its environment". The explanations contain a lot of seemingly irrelevant detail, and after reading them I'm still not sure how the SRD method works in general. It would be helpful if the authors could add an algorithm box with some pseudocode for the method.

It seems clear that the proposed method would not scale to more complex environments. The neural networks for the two environments seem to be hand-designed by the authors, and it's not clear how they propose to design a network for a new problem. The paper seems to assume an "imperfect designer" who makes mistakes in the reward function but at the same time is is willing to put a lot of effort into hand-designing a neural network. I think improving the reward function would be much easier for the designer than hand-designing the network as proposed in this paper.

The paper makes the following strong claim: "The fact that solutions can be achieved by our NN-based models *even without training* means that meaningful and interpretable weights can be purposefully designed to solve problems with more transparency." Since the method was only demonstrated on gridworlds, this claim is not well-supported by the paper.

The paper also claims that the proposed neural network designs are "highly interpretable" but this claim is not well-supported. The experimental section shows various sample trajectories but does not use the interpretable design to explain why the robot follows these trajectories. In fact, the paper suggests that the network weights do not explain the failure modes of the policy: "We see that even failure modes can yield weights profile that look similar to non-failure modes". The collection of experiments is not well-motivated (with unintuitive names like "Compare A") and it was not clear to me how they demonstrate interpretability.

The paper does not compare the proposed method to any baselines, e.g. any of the related work discussed in the introduction, so the paper does not demonstrate an advantage of the proposed method over other methods.

**Summary Of The Paper:**

The paper proposes the Self-Reward Design method for learning a reward function and designing an interpretable neural network for a specific task. It illustrates the method on two gridworld environments, RobotFish and LavaLand, using a hand-designed network for each environment.

**Summary Of The Review:**

This paper is poorly written and the contributions do not seem significant, so I cannot recommend acceptance for this paper.

---

> ### Author Response · Authors · 2021-11-09
> **Author's Response**
>
> Thank you for your feedback! We will revise for better readability. Regardless, we will try to address some of your concerns.
>
> **1. Comment**: The general form of the SRD method (independent of the task) is not explained in the paper.
>
> *Response*: there should not be a general form, since interpretability in our context is dependent on the task.
>
> **2. Comment**: e.g. I didn't understand what sentences like this mean: “the prefrontal cortex (PFC) modules judge the worth of the fish’s own action against its environment”.
>
> *Response*: if you look closely, this is the verbal description of equation (2).
>
> **3. Comment**: It seems clear that the proposed method would not scale to more complex environments. The neural networks for the two environments seem to be hand-designed by the authors, and it's not clear how they propose to design a network for a new problem.
>
> *Response*: We show that it is possible to thoughtfully consider the problem (robot fish, grid world etc) in setting up the neural networks, even down to initializing weights. New problem can be considered in similar ways.
>
> **4. Comment**:  The paper seems to assume an "imperfect designer"
>
> *Response*: this is a valid problem brought up by the paper Inverse Reward Design we cited.
>
> **5. Comment**:  I think improving the reward function would be much easier for the designer than hand-designing the network as proposed in this paper.
>
> *Response*: perhaps *easier* is better, if the ease is the main concern rather than interpretability. But this is going off our main focus entirely, which is to demonstrate the possibility of highly human-centric design.
>
> **6. Comment**: “The paper makes the following strong claim: … Since the method was only demonstrated on gridworlds, this claim is not well-supported by the paper”.
>
> *Response*: very good point, we must admit. But if this paper is refused on that ground and no further attempts to try solving problems the way we try here, probably we are closing the door to the possibility we demonstrate here: designing with meaningful and interpretable weights.
>
> **7. Comment**: The paper also claims that the proposed neural network designs are "highly interpretable" but this claim is not well-supported.
>
> *Response*: robot fish example should be clear enough since we have a meaning for each neuron. For grid world robot, the interpretability should be clear in figure 2 through the colour map in $v_1$ e.g. red is the desirable position, such as the target. The final decision of the robot depends on the total sum of the $v_{\Sigma}$ along the different trajectories. We are not sure how much clearer it should be.
>
> **7. Comment**: In fact, the paper suggests that the network weights do not explain the failure modes of the policy.
>
> *Response*: yes, we cannot explain them fully yet. We suspect it could be the artifact of our random choice that allows the robot to explore. We leave this for future studies since we do not believe the overall study of stability can be put aside with a simple paragraph.
>
> **8. Comment**: The paper does not compare the proposed method to any baselines,
>
> *Response*: baseline in problem-solving accuracy could be done, but it can distracting us away from our focus: interpretable human design. we believe achieving ~90% accuracy is reasonably high since we allow 10% randomness for exploration.
>
> **9. Comment**:  The collection of experiments is not well-motivated (with unintuitive names like "Compare A") and it was not clear to me how they demonstrate interpretability.
>
> *Response*: we start with Project A. Then we do Compare A which is similar to Project A, but with changes in parameter. The rest are similar.

---

> > ### Comment · Reviewer_cYQC · 2021-11-15
> > **Main concerns not addressed**
> >
> > Thank you for revising the paper, which is now somewhat easier to read. Unfortunately, my main concerns with the paper have not been addressed.
> >
> > 1. If there is no general form for the SRD method, how can other researchers apply it to new problems besides the gridworlds in your paper? Based on a quick look at the revised paper, I still don't know what the proposed method is, let alone how it might be applied in a new context.
> > 2. Sure - my point was that there were a lot of similar sentences in the paper that were hard to understand.
> > 3. The kind of thoughtful analysis of the problem and manual design of the neural network structure and weights that you perform on the small gridworld environments does not seem feasible for more complex environments.
> > 4. My issue is not with the assumption that the designer is imperfect, it's the inconsistency with the assumption that the designer is willing to do a lot of work to construct the neural network.
> > 5. Ease of use for the method is always a consideration - if your method is too laborious, it will not be useful. Considering the tradeoff between interpretability and ease of design is not optional - this is a key challenge of the interpretability field. The possibility of highly interpretable design at any cost seems obvious and not very useful to demonstrate.
> > 6. I think the door on this possibility is already closed for more complex problems, because the method does not scale.
> > 7. My issue here is that you seem to define "interpretability" as giving intuitive names to different nodes in the network. This is different from how I understand the usual definition of interpretability, which has to do with explaining why the agent acts the way it acts and predicting how it might act in new situations. For example, I think that explanation of failure modes of the agent is definitely within the scope of an interpretability paper and should not be left to "future studies". I don't see how your method is helpful for the purpose of explaining and predicting the agent's behavior, and so I think that it's somewhat misleading to claim that you method is "highly interpretable".
> > 8. By baselines, I meant other interpretable RL methods, to support the claim that your method is more interpretable than others.
> > 9. The revised version of the paper still has the unintuitive names for different experiments. I think this creates a lot of work for the reader to understand how the experiments differ from each other and what they demonstrate. For example, what does "A" stand for in these project names?

---

> > > ### Author Response · Authors · 2021-11-24
> > > **Thank you again for your comments**
> > >
> > > The feedback is very much appreciated and we will consider them in the near future.
> > >
> > > Some conflicting points that we believe are irresolvable:
> > >
> > > 1. Definition of interpretability. We are actually just surprised that there is still an issue with interpretability even when we attempt to construct network with such a detailed and fine-grained bottom-up piece-by-piece set up. We have no argument against that, to be honest.
> > >
> > > 2.  The need for general form. This is probably tied to "ease of use" concern. We opt to sacrifice ease of use for interpretability. The end point is clear: we want to attain a very high level of transparency. If interpretability is not worth the human effort "to do a lot of work to construct the neural network", then, I suppose the whole interpretability community could continue with high level heuristics that sometimes hardly explain anything.
> > >
> > > 3. Explanation of failure modes. For now we can only say that failure mode stems from randomness, and randomness sometimes leads us to an extreme state. To try to explain in a fine-grained way is to try to include explanation on randomness, which we don't think anyone can do. A statistical analysis may be useful, but that is again beside the point.
> > >
> > > Note: project A is the first in the series of experiment. We also include project B, C, D which are all the same as A, for reproducibility purposes. We make sure that the first result we get is not obtained by pure chance. This is mentioned in the paper.

---

### Official Review · Reviewer_NCWP · 2021-10-30

**Correctness:** 2
**Technical Novelty And Significance:** 2
**Empirical Novelty And Significance:** 2
**Recommendation:** 3
**Confidence:** 2

**Main Review:**

This paper proposes an interesting idea of designing specialized neural networks for maximum interpretability, so that the human designer can inject as much semantic information in the feature space as possible to simplify learning. Explainable machine learning is an important topic of research and designing ways to allow humans to inject as much domain knowledge as possible for learning agents is also an important direction for allowing machines to quickly acquire knowledge from their human teachers. However, the method proposed in this paper seems a bit too ad-hoc for specific problems and grants no generalizability to complex real world problems. The two task domains used in this paper are too simple to demonstrate the usefulness of the proposed idea. The writing of this paper is also not very professional and was hard to follow.


**Summary Of The Paper:**

This paper proposed to design specialized neural networks for different task domains such that the feature space is fully interpretable by human domain experts and therefore offers the opportunity for them to inject semantic biases.

**Summary Of The Review:**

This paper proposes a novel idea for building explainable neural networks but lacks strong theoretical and empirical evidence to support its claim. I do not recommend accepting this paper.

---

> ### Author Response · Authors · 2021-11-09
> **Author's Response**
>
> Thank you for your feedback! We will revise for better readability. Regardless, we will try to address some of your concerns.
>
> **Comment**: the method proposed in this paper seems a bit too ad-hoc for specific problems and grants no generalizability to complex real world problems
>
> *Response*: similar to review by cYQC, see our response to comment 3.

---

### Official Review · Reviewer_sPTc · 2021-11-01

**Correctness:** 2
**Technical Novelty And Significance:** 2
**Empirical Novelty And Significance:** 1
**Recommendation:** 1
**Confidence:** 4

**Main Review:**

-The motivation of this work is RL and DL, but the proposed method cannot be used to solve the problems those approach can solve. The connections to RL are explained in an unclear fashion.

-"Trading off" "time spent on training algorithm with the time spent on human design" is at odds with how RL is used. Basic RL can be used without a domain model, human expertise, etc. If a human is available to hand-specify a policy, other approaches are generally preferred. This work is not compared to other approaches for hand-specifying policies.

-After manually specifying parameters, training is performed. It is unclear why this training would preserve the meaning of each neuron.

-The interpretability of the proposed system is not demonstrated.

-The system is changed for the different domains, and different means of explaining it are used. It is unclear how the proposed method would be applied to further domains (since two special cases were presented).

Minor Comments:

-Splitting the Introduction and Related Work would improve clarity.

-This work would benefit from another editing pass.

**Summary Of The Paper:**

To create an interpretable system, the authors propose to manually construct neural networks such that each neuron corresponds to an interpretable concept. Such a system can then be trained to improve its performance.

**Summary Of The Review:**

1. This work does not present a systematic approach that can be broadly used.

2. The interpretability of the final model is not shown.

3. This work attempts to make connections to DRL, but the proposed method is not applicable to the same problems.

---

> ### Author Response · Authors · 2021-11-09
> **Author's Response**
>
> Thank you for your feedback! We will revise for better readability. Regardless, we will try to address some of your concerns.
>
> **1. Comment**: “The connections to RL are explained in an unclear fashion”.
>
> *Response*: we pointed out in the paper: “While we try to incorporate interpretability, we have diverted away from the standard RL”, p.2. Our model resembles RL only to limited extent, and their connection does not need to be asserted. We do not require ourselves to follow the RL trend strictly.
>
> **2. Comment**: “This work is not compared to other approaches for hand-specifying policies”
>
> *Response*: The context is necessary. RL community has gone from standard RL to deep RL. The problem of interpretability arises in deep RL because of the use of black-box deep neural network (DNN). We are trying to solve the latter problem, by using DNN components, but we use them in interpretable ways.
>
> **3. Comment**: "Trading off" "time spent on training algorithm with the time spent on human design" is at odds with how RL is used
>
> *Response*: exactly. We are trying to suggest something that avoids excessive black-box DNN training (see response to comment 2) and make something that is human-centric.
>
>
> **4. Comment**: “After manually specifying parameters, training is performed. It is unclear why this training would preserve the meaning of each neuron”
>
> *Response*: we can actually see from fig. 5 that the weights specified with high center value is preserved. This is the interpretability we initially want, and the property is preserved after training.
>
> **5. Comment**: “The system is changed for the different domains, and different means of explaining it are used.”
>
> *Response*: similar to review by cYQC, see our response to comment 3.

---

> > ### Comment · Reviewer_sPTc · 2021-11-19
> > **Response to authors**
> >
> > Thank you for the commitment to address readability concerns.
> >
> > However, the core concerns still remain. I largely agree with reviewer cYQC, and the comments generally do not resolve this work's shortcomings.

---

### Official Review · Reviewer_NiCV · 2021-11-01

**Correctness:** 2
**Technical Novelty And Significance:** 2
**Empirical Novelty And Significance:** Not applicable
**Recommendation:** 3
**Confidence:** 4

**Main Review:**

Interpretability for deep reinforcement learning is an important and significant topic of research. This paper suggests a mechanism that aims to combine the benefits of human interpretable models with the performance of methods that leverage gradient based optimization. However, the paper does not provide sufficient evidence, empirical or theoretical, to justify the use of these hand-designed neural networks. Some of the major concerns with the paper are:
1. The mechanism is not scalable. Designing individual neurons of a network will likely become intractable for even reasonably complicated tasks.
2. There are no comparisons with existing techniques that aim to learn interpretable policies.
3. The optimization techniques does not seem to provide a substantial performance improvement, and in some cases degrades the performance of the agent.

The lack of improvement via optimization is particularly significant since in the absence of gradient based improvements there seem to be no reason to use this mechanism.

**Summary Of The Paper:**

This paper proposes using individually designed, interpretable neural networks to solve a given task. Each neuron in the network is hand-designed to serve a specific task-dependent purpose. These hand-designed solutions are then optimized using environment interactions. The neuron level design of the network is used to provide human interpretability, as the behavior of the network can be understood based on the activations during environment interactions. The neural network structure of the policy can then be optimized via gradient descent to provide a performance improvement over the hand-designed solution.

**Summary Of The Review:**

The shortcoming noted in the main review suggest that the proposed mechanism for generating interpretable policies is unlikely to provide a viable alternative to previously proposed methods. Considerable improvements would be required to meet the threshold for acceptance.

---

> ### Author Response · Authors · 2021-11-09
> **Author's Response**
>
> Thank you for your feedback! We will revise for better readability. Regardless, we will try to address some of your concerns.
>
> **Comment**: The mechanism is not scalable.
>
> *Response*: similar to review by cYQC, see our response to comment 3.
>
> **Comment**: There are no comparisons with existing techniques that aim to learn interpretable policies
>
> *Response*: Comparison for interpretability? We believe it is best to show the interpretability as clearly as it can be, e.g. through naming each neuron meaningful (fig. 1) and through tile-based red/blue scores (fig. 2). While other papers have claimed interpretability through other means, our design is directly manipulating the weights and biases, i.e. we handle the interpretability down to the most fine-grained aspect of the architecture. As for accuracy comparison, we believe achieving ~90% accuracy is reasonably high since we allow 10% randomness for exploration.
>
> **Comment**: The optimization techniques does not seem to provide a substantial performance improvement and in some cases degrades the performance of the agent.
>
> *Response*: it is still a demonstration that optimization can augment the performance due to human design. Degradation of performance needs to be considered in a separate further study on stability, which is beyond the scope of this paper.

---

> > ### Comment · Reviewer_NiCV · 2021-11-09
> > **Core issues not addressed**
> >
> > 1. The mechanism is not scalable.
> >
> > It seems the response here is that neural networks can be hand-designed for individual problems in a similar way to the current examples in the paper (robotfish, gridworld). However, it is unclear that this can be done for reasonably complicated environments with higher dimensional state spaces. To substantially address this weakness, the paper should present an algorithm that can be applied to a wide variety of environments and show (empirically or theoretically) that the algorithm scales reasonably as the state space dimension increases.
> >
> > 2. There are no comparisons with existing techniques that aim to learn interpretable policies
> >
> > While it is true that this paper proposes providing interpretability via direct manipulation of weights and biases, this in itself is not sufficient to demonstrate the value of the proposed method. In particular, the reader needs to know why they might choose this method instead of other methods that provide comparable interpretability. This knowledge needs to be provided via a through comparison against existing approaches that generate interpretable policies.
> >
> > 3. The optimization techniques ... degrades the performance of the agent.
> >
> > I respectfully disagree with the authors, this issue is very much within the scope of this paper. The proposed optimization technique is a key component of the contributions of this paper, and its effects on a policy's performance need to be studied carefully to judge whether the contributions are significant.
> >
> > Based on the above reasons, my score remains unchanged.

---

> > > ### Author Response · Authors · 2021-11-10
> > > **What we will revise on**
> > >
> > > Thanks for sparing more time for further review, much appreciated! Do not worry about the score, by the total scores, this paper is rejected anyway.
> > >
> > > **Comment 1**. The mechanism is not scalable.
> > >
> > > We will rewrite the paper so that it focuses on marketing the bits and pieces of DNN modules that are designed to tackle problem in a humanly-interpretable and meaningful way. For more complex system, we hope that readers in the future will look at how we assemble our pieces, and probably come up with the piece of modules relevant to their problems.
> > >
> > > **Comment 2**. There are no comparisons with existing techniques that aim to learn interpretable policies
> > >
> > > We focus *very very very heavily* on manual human-interpretable design. Existing interpretable RL methods can talk about regularizations, heuristics and what not, but, at the end of the day, what can we say? Our weight/bias manipulation is already at an extreme side of interpretability while existing works that we cite like (Zambaldi et al. (2019)), (Juozapaitis et al. (2019)) etc are high level analysis or implementation. As much as we want to compare them,  we can only assert that our design touches the lowest level of abstraction and tries to incorporate human design from there. Oh, yes, we will include this in our revision anyway.
> > >
> > > **Comment 3**. The optimization techniques ... degrades the performance of the agent.
> > >
> > > We would like to point out that stability analysis is indeed beyond the scope of the paper. It is clear enough that instability occurs stochastically, and, for this interpretable RL-like paper, we might need a whole series of probability methods to pinpoint the cause of instability. We will write this into our revision.

---

### Official Review · Reviewer_cBTu · 2021-11-02

**Correctness:** 1
**Technical Novelty And Significance:** 1
**Empirical Novelty And Significance:** 1
**Recommendation:** 1
**Confidence:** 4

**Main Review:**

Strengths:
- Throughout the paper, the only thing that makes sense in my opinion is that the goal of incorporating human knowledge into the network design.

Weaknesses:
- Terrible writing. Really hard for readers to understand. Lots of unexplained notations and technical details but much less illustration of the intuition.
- Unreasonable technical approach. Why does the network need to be trained if the goal is just mimicking the greedy action? Can we just use the greedy output instead?
- Hard to generalize. The network is designed case by case with heavy human effort, I don't see any learnable component in the network that makes sense and effectively supplements the human design.
- Unconvincing results. This is probably because the technical approach is not reasonable. SRD fails to improve the performance in general by looking at the result figures.

**Summary Of The Paper:**

This paper proposes incorporating (a large amount of) human knowledge into policy network design such that it solves the problem directly or within a few iterations of training. The network is trained by the proposed Self Reward Design (SRD) mechanism to simply consolidate its greedy decision, which as claimed by the authors would help supplement the human design of the network.

**Summary Of The Review:**

A clear rejection based on the weaknesses mentioned above and no intuition to the community.

---

> ### Author Response · Authors · 2021-11-09
> **Author's Response**
>
> Thank you for your feedback! We will revise for better readability. Regardless, we will try to address some of your concerns.
>
> **1. Comment**: Terrible writing. Really hard for readers to understand.
>
> *Response*: we are very sorry about that and will try to make a major revision that emphasizes on clarity.
>
> **2. Comment**: Hard to generalize. The network is designed case by case with heavy human effort.
>
> *Response*: human effort is the whole point in the context of our interpretability. If the paper is read with a preconceived expectation of “achieving good accuracy”, then yes, this paper is nearly pointless (except it does performs reasonably on the task).
>
> **3. Comment**: Why does the network need to be trained if the goal is just mimicking the greedy action?
>
> *Response*: take deep Q learning as comparison. The Q value is trained so that agent can greedily choose the greedy action in that state. But what is missing? The interpretability, because of the black-box. Deep Q Network can tell us which greedy action to take, but not why. In our model, we see a meaningful contribution that directly corresponds to what we can see: the value contributed by each tile. The reward is decomposable, in a sense. Remember, our focus is interpretability.
>
>
> **4. Comment**: I don't see any learnable component in the network that makes sense and effectively supplements the human design.
>
> *Response*: really sorry, but what sense are you referring to? In robot fish, the neuron component even has a name, if it helps. We have e.g. FC (fully connected layer) and FLD (made up of convolutional layers) as learnable components, just to mention them explicitly for the benefit of readers who are not familiar with the components in deep neural networks.  Likewise, in robot 2D, the name of each tile is also available, and their contribution to the ‘reward’ is clearly implemented through $DS_{tile}^5$.
>
> **5. Comment**: SRD fails to improve the performance in general by looking at the result figures.
>
> *Response*: this is simply an inaccurate statement. Generally, accuracy is improved for Project A, but not Compare A which uses a more extreme parameter (for comparison).

---

### Author Response · Authors · 2021-11-15
**Major Reorganization of Papers (Part 1)**


We thank all reviewers for the helpful reviews. The following is the summary of our revision. Please do refer to the reviewers’ comments for more remarks and our targeted responses.

## Major Edits

**Generalization and Scalability**. Most reviewers mention that our methods are not scalable to complex problems, which we have admitted as our limitations before revision. In our revised paper, we include additional remark to encourage users to apply highly interpretable design: “Future works on applying interpretable SRD on complex systems might push Deep RL community into designing systems that aim not only to attain high performance, but also to be as highly interpretable as our design”.

**Paper structure**. We thank reviewers for relevant comments on the clarity and presentation of paper e.g. (1) Reviewer cBTu comment “Lots of unexplained notations and technical details but much less illustration of the intuition”, (2) reviewer sPTC’s minor comments and (3) reviewer NCWP’s comment “The writing of this paper is also not very professional and was hard to follow”. We edit the presentation of our idea in response to comments regarding clarity. *Now the narrative of the paper is extremely linear*, as the following.
In summary, section 2 now is arranged as the following with clearly named sections:
1. **What exactly is this paper about?** Interpretable design to RL problem. This is for reviewers who find the paper hard to read. We also mention “Readers will find how we design different components taylored to different problem components”.
2. **What is interpretability?** Direct weight and bias manipulation. This is also for reviewers who find the paper hard to read, e.g. (1) reviewer sPTc who mentions that ‘The interpretability of the proposed system is not demonstrated’ and (2) reviewer cYQC who comments that ‘proposed neural network designs are "highly interpretable" but this claim is not well-supported’.
3. **How to compare interpretability?** This is in response reviewers’ comment on the lack of comparison e.g. Reviewer NiCV comment 3. We assure that our interpretability is almost maximum, as mentioned by Reviewer NCWP.
4. **Baseline**. We mention that our model achieves 90% performance given 10% random probability for exploration, thus we do not compare *accuracy* performance with existing models.
5. **Different design for different contexts**. This is to reiterate that our human design is highly interpretable and context dependent, hence different design is required for different problems.
6. The following section is the “related work”.

The rest of the paper is mostly reorganized such that each interpretable component and result is presented step by step, paragraph by paragraph in the clearest, straight-forward manner, as the following.

In summary, section 3 (Robot Fish) now is presented linearly step by step, also with clearly named sections:
1.	Problem setting. Define the environment where food is sometimes available. Now the illustration for environment is presented directly in fig. 1.
2.	Fish1D’s Actions. We clearly specify the two actions and the effects.
3.	How to design an interpretable component of neural network? This section explicitly tells readers about the neurons that are explicitly named for interpretable design.
4.	The fh and ft neurons. This is the technical details of the previous points.
5.	Interpretable FishNN. This section aggregates the neural network that makes the decision.
6.	Is FishNN’s decision correct? This section defines the architecture PFC that is used for self-rewarding. We mention that it is done the same way as FishNN is made. In particular, we made it clear that PFC is used to check if FishNN’s decision is considered true or false. This is particularly in response to reviewer cYQC’s comment “I didn't understand what sentences like this mean: "the prefrontal cortex (PFC) modules…”
7.	Self-reward optimization. We add a remark “The ground-truth used in the loss is $argmax(z)$, computed by the fish itself: hence, self-reward design”.
8.	Results.

---

### Author Response · Authors · 2021-11-15
**Major Reorganization of Papers (Part 2)**

Continued from Part 1:

In summary, section 4 (lavaland problem) is similarly presented:
1.	The lavaland problem, the problem which IRD paper used.
2.	Interpretable design. In this paragraph, we clearly state that the focus is the ‘favourability gradient’. This is in response to reviewer’s comments that interpretability is not clearly demonstrated, e.g. reviewer sPTC and reviewer cYQC.
3.	Notations.
4.	Interpretable tile-based modules. The modules demonstrate our use of neuron-wise strong activation for interpretability.
5.	Unknown avoidance. Large part of our paper tries to address the problem that Dylan’s IRD solved: imperfect designer. Our unknown avoidance does it by aggregating “the negation of known activations”.
6.	Interpretable tile scores. This paragraph specifically defines how we manually manipulate the weights and biases of the components $DS_t^5$. This is our attempt to maximize interpretability. For clarity, we include the fig. 5 showing the explicit weights here.
7.	Making one plan. This details the method on how a trajectory is made. The 10% randomness we mentioned earlier is included here.
8.	Imagining multiple plans for SRD optimization. This demonstrates how different loss is used based on different context to maintain the interpretability.

In summary, the results are now presented clearly, based on for example reviewer cYQC’s comments that “The collection of experiments is not well-motivated (with unintuitive names like "Compare A")”.
1.	Experimental setup and comparison. This includes a short remark on the names “project A” etc. In fact, we refer reader to the appendix that shortly present project B, C, D etc.
2.	Reaching targets without training, optimized by training. Clearly and briefly comparing the model with and without SRD training.
3.	Then project A, Compare A, Lava A and Lava NOAV A are presented with succinct descriptions that clearly highlight the differences. There will not be extra, unnecessary details.

## Minor changes
Hyperlink is included. Supplementary materials are updated.

---

### Decision · Program_Chairs · 2022-01-20

**Decision:**

Reject

**Comment:**

The paper proposes a design of interpretable neural networks where each neuron is hand-designed to serve a task-specific role, and the network weights can be optimized via a few interactions with the environment. The reviewers acknowledged that the interpretability of neural networks is an important research direction. However, the reviewers pointed out several weaknesses in the paper, and there was a clear consensus that the work is not ready for publication. The reviewers have provided detailed and constructive feedback to the authors. We hope that the authors can incorporate this feedback when preparing future revisions of the paper.